# Evaluation of the virtual care experience for persons in prospective cohorts with HIV during the COVID pandemic

Sharon L. Walmsley[1]*, Majid Nabipoor[2], Valerie Martel-Laferriere[3], Mona Loutfy[4], Curtis Cooper[5], Marie-Louise Vachon[6], Bryan Boyachuk[1], Pamela Aldebes[7], Marina B. Klein[7], CTN-COVID Sub-study Group

1 University of Toronto, Infectious Diseases, Toronto, Canada, 2 University Health Network, The Biostatistics Research Unit (BRU), Toronto, Canada, 3 CHUM - Centre Hospitalier de l'Université de Montréal, Montreal, Canada, 4 Women's College Hospital Research Institute, Toronto, Canada, 5 The Ottawa Hospital, Ottawa, Canada, 6 CHU de Québec-Université Laval, Quebec, Canada, 7 Research Institute of the MUHC, Montreal, Canada

Membership of the CTN-COVID Sub-Study Group is provided in the Acknowledgements section.
* Sharon.walmsley@uhn.ca

## Abstract

The COVID pandemic necessitated shifting to virtual care. Our aim was to describe, and identify the challenges and satisfaction with the virtual care experience of a subset of participants from two established Canadian Trials Network (CTN) cohorts: CTN 222 (HIV/HCV coinfection) and CTN 314: CHANGE HIV (Correlates of Healthy Aging in geriatric HIV infection) - persons > 65 years age. We hypothesized that vulnerable populations could face challenges with virtual care related to age, mental health or drug addiction. Consenting participants provided demographic information, completed a non-validated 18-item self- administered questionnaire on their virtual care experience, and reported HIV specific laboratory collection and prescription refills during the COVID pandemic. Data on CD4 T lymphocyte counts and HIV viral loads were extracted from medical records. A total of 454 individuals participated between February 2021 and March 2023, including 133 from CTN 314 and 321 from CTN 222. Overall, 55.3% engaged in virtual care. In multivariable regression models (analysis with SAS and R software) use of virtual care was higher in the aging cohort (p < .0001) but did not vary with current alcohol, drug use or self-reported depression (p > .05). The most common reason for not engaging was that it was failure to offer. Of those who engaged, 55% reporting being very satisfied, 36.3% somewhat satisfied, and 8.8% not satisfied. Ten percent of the older and 16% of the HCV cohort, reported technology difficulties as a barrier to use. Those with a detectable HIV viral load were more likely to engage in virtual care, p < .05. 81.3% of participants had HIV blood tests as frequently as before the COVID-19 pandemic. Despite high satisfaction, the majority (80%) prefers in person visits. When offering virtual care, clinics need to ensure all eligible patients are aware of how to access the services and consider patient needs and preferences.

**Data availability statement:** Data provided as a supplement.

**Funding:** This work was supported by a research grant from the Investigator-Initiated Studies Program of Merck Canada Inc (to SW, MK). The opinions expressed in this paper are those of the authors and do not necessarily represent those of Merck Canada Inc. The funders had no role in study design, data collection and analysis, decision to publish or preparation of the manuscript.

**Competing interests:** The authors have declared that no competing interests exist.

## Author summary

With the onset of the COVID pandemic, much of outpatient health care had to shift to a virtual format either audio or video. We had concerns that our population living with HIV would have challenges with this technology and that it might be particularly difficult for our vulnerable populations such as those with hepatitis C coinfection who frequently have mental health issues and drug addiction and those who are aging. To evaluate this, we surveyed persons living with HIV who had been enrolled in two prospective cohorts. They completed an 18-item survey about their virtual care experience. We discovered that about half had engaged in virtual care, more in the older cohort than in the HCV cohort. The main reason for not engaging was that virtual care had not been offered to them. Although most who did engage in virtual care were satisfied with their experience, the majority preferred in person visits. We conclude that there are advantages and disadvantages of virtual care but that clinics need to be flexible and provide the method of care tailored to the individual. If virtual care is to be offered patients need to be informed of the means by which to access it.

## Introduction

During the COVID pandemic many aspects of general and specialty outpatient medical care transitioned to either audio or video virtual appointments [1,2]. While care providers can evaluate symptoms and provide emotional support virtually, many aspects of physical examination are not possible. Many physician offices and hospital clinics struggled with prioritizing patients for in person assessment [2].

As the population of persons living with HIV in Canada and elsewhere are more economically disadvantaged than the general population, there were further concerns about access to devices and internet connections needed for virtual care. For example, in a study of 1403 women living with HIV in Canada, 65% reported an income of less than $20,000 per year which could limit their ability to own devices [3]. For persons living with HIV there are other considerations including privacy in the home, at work or on the street when attending virtual appointments. Given the stigma that remains around an HIV diagnosis, persons may not have previously disclosed their status to others in the home for fear of violence or social non-support [4]. With fear of contracting COVID-19 and the disruption of laboratory services, there was further concern regarding delays in regular laboratory testing with negative consequences including missed adverse drug effects, failure to detect and intervene on HIV viral failure or non-adherence and undetected sexually transmitted infections [5]. In contrast, others proposed virtual care represented a new care model for persons living with HIV, particularly for those in good health with well-controlled infection, enabling them to continue gainful employment without the need for time away from work for medical appointments. Potential advantage was also identified for those with mobility issues or those living in rural areas with less access to specialty care [1].

The advantages and disadvantages of virtual care may vary by population characteristics. Those who are aging, have co-morbidity, face mental health challenges, live with drug addiction or are homeless could encounter more challenges with virtual care. For such people, personal connections with their health care team are key to successful therapy and likely disrupted without face-to-face encounters.

We evaluated the virtual care experience of persons living with HIV from two established cohorts: Canadian HIV trials network- CTN 222 -a HIV-HCV co-infection cohort [6] and CTN 314- CHANGE HIV- a cohort of persons with HIV aged 65 years or older [7].

We hypothesized that the elder population may face technology challenges in accessing virtual care, while those with HCV who have more issues related to mental health, or drug use may not have the resources for devices or internet access to participate in virtual care. These vulnerable groups may also have had more concerns about attending in person due to the risk of COVID exposure yet addressing their health issues could be challenged by virtual appointments.

## Methods

### Ethics statement

This sub-study was approved by the University Health Network Research ethics board (CAPCR # 20–5986) and by the REB of the participating sites. All participants gave written informed consent. Both parent studies had previous approval by their local REB and the CTN community advisory committees and received approval for the transition to online assessment.

### Protocol

At the onset of the HIV pandemic, CTN affiliated cohort studies were considered as non-essential research and onsite visits for research purposes were not permitted. Two parent cohort studies- CTN 222 and CTN 314 pivoted to on-line methods and questionnaires for follow up. The objectives and demographics of the separate studies are published and are available on their respective websites [6,7] (https://www.changehivstudy.com/; https://cocostudy.ca/). As the parent studies were transitioning, we added this optional sub-study to address the impact of the COVID pandemic on physical and emotional health, and to assess the virtual care (either audio or video) experience.

Active participants of the two cohorts were approached at their follow up visit for the main study and with consent, completed a questionnaire on the virtual care experience, HIV specific laboratory testing and medication adherence. Demographic data, substance use, and current depression were self-reported and most recent T lymphocyte count (CD4) and viral load information (prior 3 months) collected from the participants medical records.

This was a cross-sectional descriptive study using a self-completed questionnaire. The Virtual Care Questionnaire we used was developed by two of the investigators (SW, MK) in collaboration with the 12-member community advisory committee of the CTN 314 study (provided in Appendix). It consists of 18 questions addressing the use of, satisfaction with and perceived strength and weakness of virtual care. The questionnaire defined virtual care as having an appointment with the doctor or other health care provider over the phone or the computer. It also addressed completion of HIV specific blood work (CD4 cell count and viral load) and medication refills. Questions were completed with single or multiple choice check boxes. This questionnaire had not been previously validated nor had reliability testing before use in this study. It was completed between February 2021 and March 2023 when participants attended their visits for the parent studies.

### Statistical analysis

Cohort characteristics and questionnaire responses were compared between the CTN 222 and CTN 314 sub-study participants.

The association of demographic characteristics- cohort, age, race, gender, income, active drug use and self-reported depression with use of virtual care, and the ability to complete HIV laboratory monitoring and antiretroviral medication refills using logistic regression. For questions with three level responses ordinal logistic regression was used for analysis.

Univariable and multivariable models were constructed. Tests of proportional odds assumptions for ordinal logistic regression were conducted using the Brandt test. A significance level of α = 0.05 was applied to all statistical hypothesis tests. Data manipulation and analyses were performed using SAS (version 9.4) and R software (version 4.2.2).

## Results

Seven of nineteen active CTN 222 sites and 2/6 active CTN 314 sites were able to participate in this sub-study. During the recruitment period (February 2021- March 2023) the virtual care questionnaire was completed by 454 participants including 133 (51%) of active participants from the contributing CTN 314 sites and 321 (76%) of active participants in the contributing CTN 222 sites.

The demographics of those participating in the sub-study by cohort are in Table 1. There are several differences in participants of the two cohorts. The median age was 71 years in the aging cohort and 45 years in the coinfection cohort. There were more females in the coinfection group (29% vs 8%). The reported income was lower in the HCV coinfection cohort where 75% reported an annual income of lower than $50K Canadian. The median duration of HIV infection was slightly longer in the older cohort (26 vs 22 years). The prevalence of HIV RNA > 50 copies/ml was higher in the coinfection cohort (13% vs 1%) whereas the CD4 counts were similar with over 50% > 500/mm3 in both groups. Self- reported depression was noted in 43% of the coinfection cohort compared to 15% of the older group. Alcohol use was similar in the two groups but substance use higher in the coinfection cohort.

Overall, 55% reported having experienced virtual care including 86.5% of those in CTN 314 and 42.4% of those in CTN 222 (Fig 1). The main reasons for not having experienced virtual care was that it had not been offered to them or they preferred to be seen in person. Of those who did experience virtual care it was most commonly with either their primary care provider or their HIV specialist.

We conducted a multivariate analysis to assess for factors associated with the use of virtual care (Fig 2). Participants from the older cohort (CTN 314) were more likely to have experienced virtual care although age alone was not predictive. Those who had a detectable viral load and lower income were also more likely to access virtual care on univariate but not multivariate assessment.

We assessed satisfaction with the virtual care experience (Fig 3). Over 55% of the participants were very satisfied with their virtual care experience and approximately 60% felt their clinical concerns had been met. Most felt it had been easy to book a virtual appointment with their health care provider and less than 15% experienced problems with the audio or the video. Levels of satisfaction were similar between participants of the two cohorts.

Overall, 21.5% of participants felt that virtual care was better than in-person care (Fig 4). The reasons cited for preferring virtual care included not needing to travel (17.9%), faster service (13.9%), feeling safer (6.4%), not missing work (4.4%), and finding it easier to discuss concerns (4.4%).

Despite the satisfaction, 70–80% preferred in person care. Overall, 67.7% of participants missed aspects of in-person care (Fig 5). This concern was more common in the older cohort compared to the HCV cohort (76.5% vs 60.3%, p = 0.0061). The specific aspects missed included missing human contact (54.6%), the absence of a physical exam (29.9%), concerns not being fully understood (13.9%), inability to do blood work (12.4%), and being unable to meet with a nurse, pharmacist, or social worker (11.2%).

Overall, 81.3% of participants had HIV blood tests as frequently as before the COVID-19 pandemic. They conducted their tests at the hospital (75.3%), test centers (17.1%), and family doctor clinics (6.5%). Among those who did not have a blood test, the reasons included: not being sure it was still possible (48.2%), being told by their doctor to hold off (27.1%), not wanting to do the test (21.2%), and not feeling safe to leave the house (16.5%).

We conducted univariate and multivariate analysis to assess for factors associate with regular blood test monitoring during the pandemic (Fig 6).

Only alcohol use was marginally associated with less routine laboratory monitoring.

**Table 1.  Demographic and baseline laboratory characteristics of study participants by cohort.**

| Characteristics | CTN222 = Canadian Co-Infection Cohort | CTN314 = CHANGE HIV (> 65years age) |
|---|---|---|
| Total, N | 321 | 133 |
| Age | | |
| Mean (SD) | 45 (10) | 70.58 (4.4) |
| Gender, n (%) | | |
| Female | 92 (28.7%) | 11 (8.3%) |
| Male | 229 (71.3%) | 122 (91.7%) |
| Race, n (%) | | |
| Caucasian | 237 (73.8%) | 105 (78.9%) |
| Indigenous | 55 (17.1%) | 0 (0.0%) |
| Asian | 4 (1.2%) | 5 (3.8%) |
| Black | 14 (4.4%) | 10 (7.5%) |
| Hispanic | 10 (3.1%) | 5 (3.8%) |
| Income, n (%) | | |
| <20K | 29 (9.0%) | 22 (16.5%) |
| 20K - 49.9K | 210 (65.4%) | 48 (36.1%) |
| 50K - 99.9K | 32 (10.0%) | 36 (27.1%) |
| >100K | 13 (4.0%) | 27 (20.3%) |
| Years with HIV | | |
| Mean (SD) | 22.45 (18.89) | 25.94 (9.00) |
| Current Alcohol use, n (%) | | |
| Yes | 160 (49.8%) | 74 (55.6%) |
| Current Marijuana use, n (%) | | |
| Yes | 172 (53.6%) | 38 (28.6%) |
| Current Cocaine/heroin use, n (%) | | |
| Yes | 110 (34.3%) | 0 (0.0%) |
| Current Opioid use, n (%) | | |
| Yes | 27 (8.7%) | 8 (6.0%) |
| Current Use of Other injectable agents, n (%) | | |
| Yes | 18 (5.6%) | 3 (2.3%) |
| Current Depression, n (%) | | |
| Yes | 137 (42.7%) | 20 (15.0%) |
| HIV viral load | | |
| >50 copies/mL | 41 (12.8%) | 1 (0.8%) |
| CD4 count/mm3 | | |
| <200 | 21 (6.5%) | 4 (3.0%) |
| 200 - 500 | 101 (31.5%) | 55 (41.4%) |
| 500+ | 159 (49.5%) | 66 (49.6%) |
| Missing | 40 (12.5%) | 8 (6.0%) |

Overall, 98% of participants refilled their HIV medications. Among the 2% who did not, reasons included being unable to pay (55.6%), lack of pharmacy delivery services (11.1%), and being too ill to visit the pharmacy (33.3%).

For the CTN 222 participants, 80.6% felt virtual care had no impact on their HCV care, 2.2% reported that their HCV treatment was affected- 80% could not start treatment, 10% ran out of medication, and 10% were unable to have a blood test.

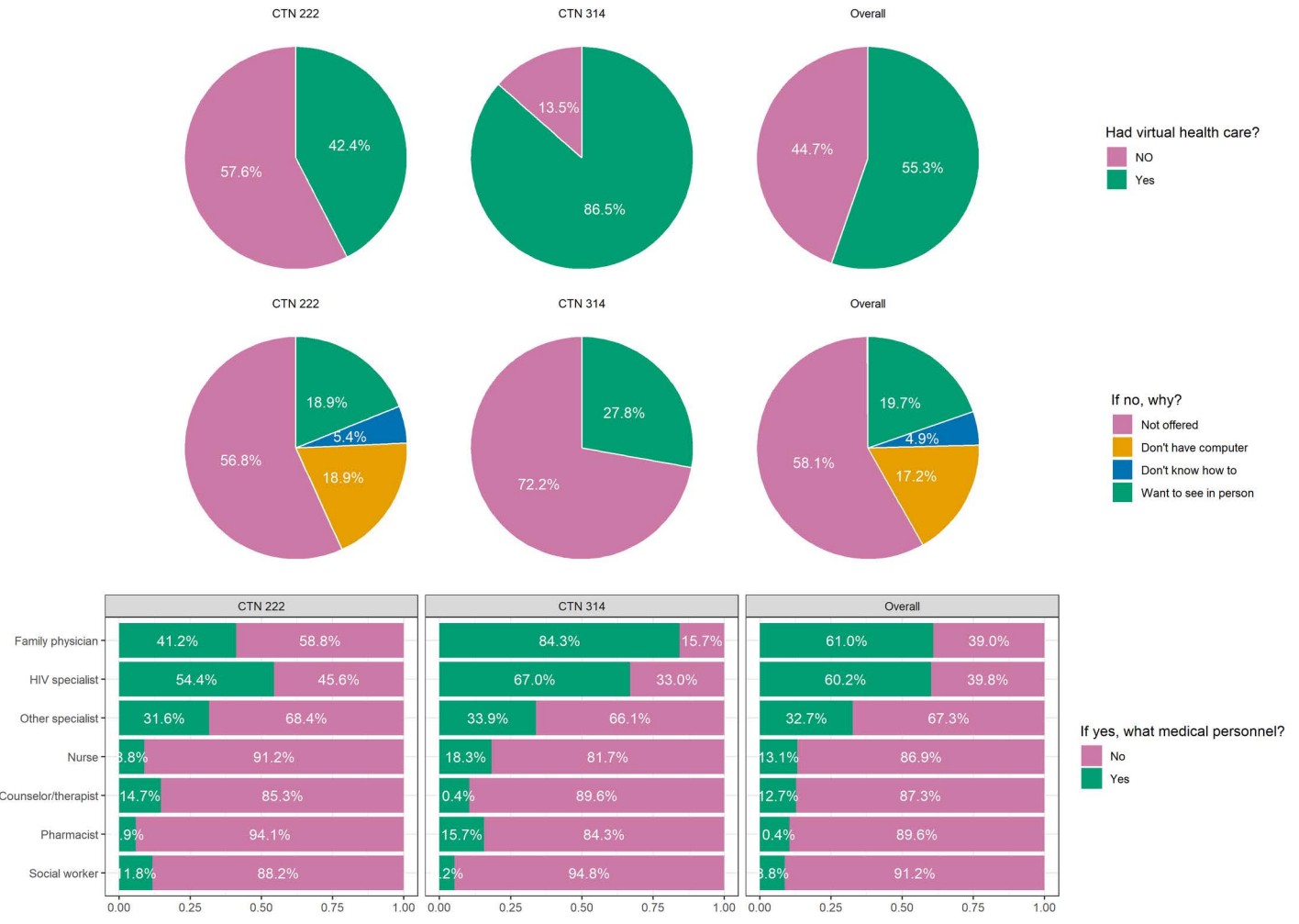

**Fig 1. Overall virtual care experience by study cohort.**

## Discussion

During the COVID pandemic, much HIV care in Canada was transitioned to a virtual platform conducted by telephone or video. We explored the virtual care experience in consenting participants of two ongoing Canadian HIV cohorts. One cohort includes only those who are aging (> 65 years) and the other to those with HIV-HCV co-infection. We anticipated that participants of these two cohorts could have challenges with use of virtual care, the older cohort because of technology challenges, and the concern about the management of comorbidity, and the HCV coinfection cohort because of lack of resources for technology and mental health and drug issues. Our results showed that overall, 55% of those participating in the sub-study had experienced virtual care including 86.5% of those in CTN 314 (older age) and 42.4% of those in CTN 222 (HCV coinfection). Others had reported that the elderly might have less access to virtual care [8]. On multivariate analysis, in our cohort increased age was marginally associated with increased virtual care use. The reason for this association is unclear. Providers may have had concerns about their older patients and may have made more efforts to ensure virtual care visits. The elderly may have had concerns about their own health and comorbidities or their risk of COVID when leaving their homes and insisted on virtual rather than in person visits. We did not find technology challenges (either access to devices or audio or technical problems) to have impaired virtual care with our older population. In contrast, the

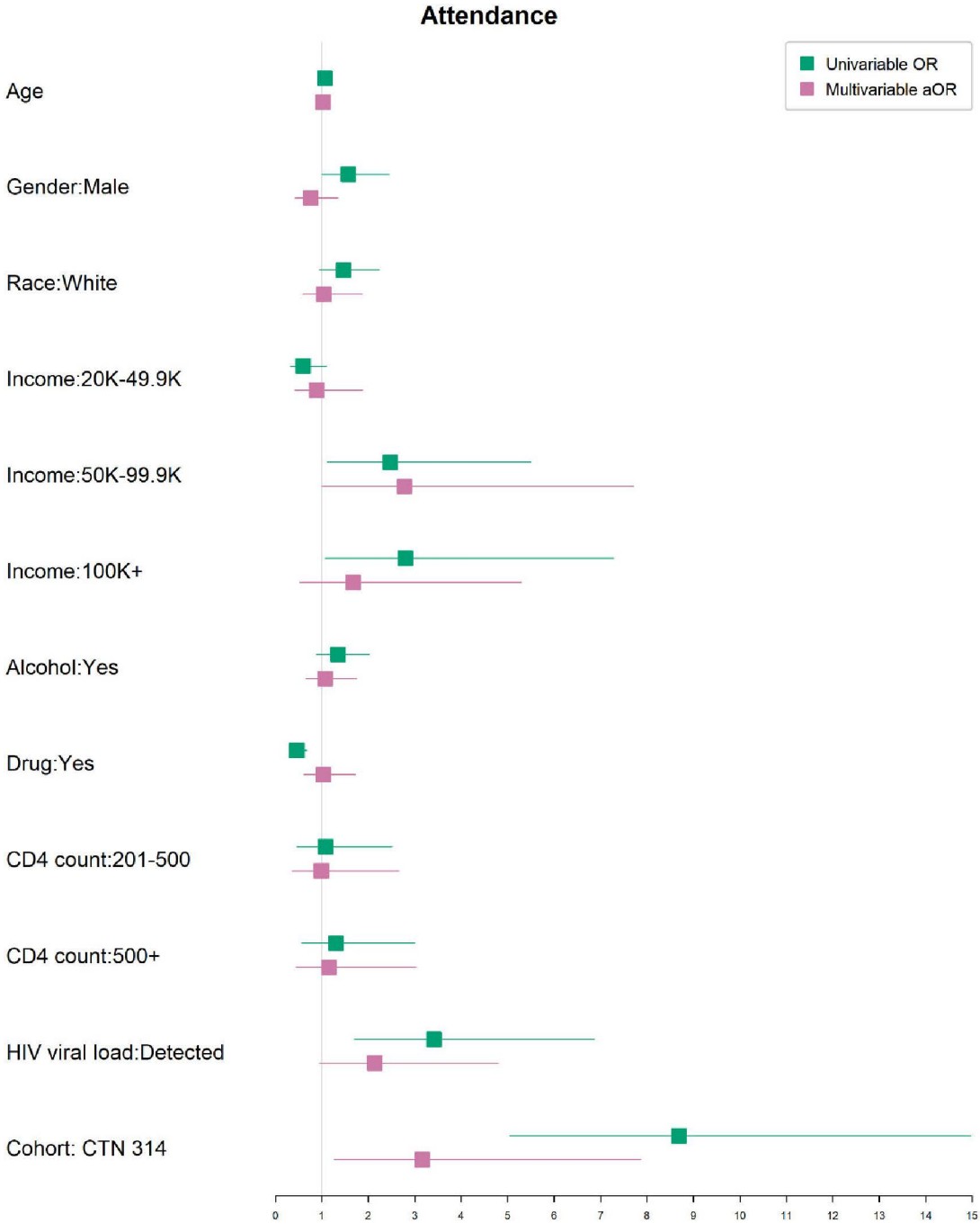

**Fig 2. Univariate and mulitivariable analyses of baseline factors associated with use of virtual care.**

HCV cohort were less likely to access virtual care. We speculate that this could be related in part by lower income, higher rates of substance use and heavier burden of depression.

The primary reason cited for not using virtual care was that it had not been offered. It is unclear whether this occurred because their particular health provider was not making this service available or whether it was because the participant

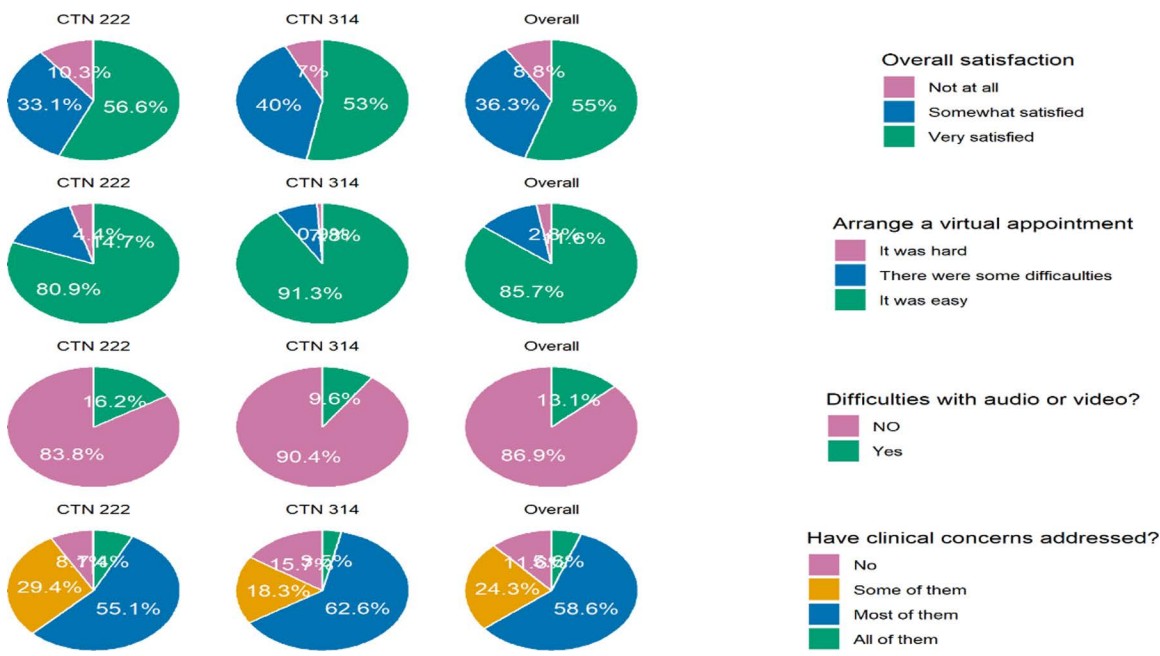

**Fig 3. Satisfaction with the virtual care experience.**

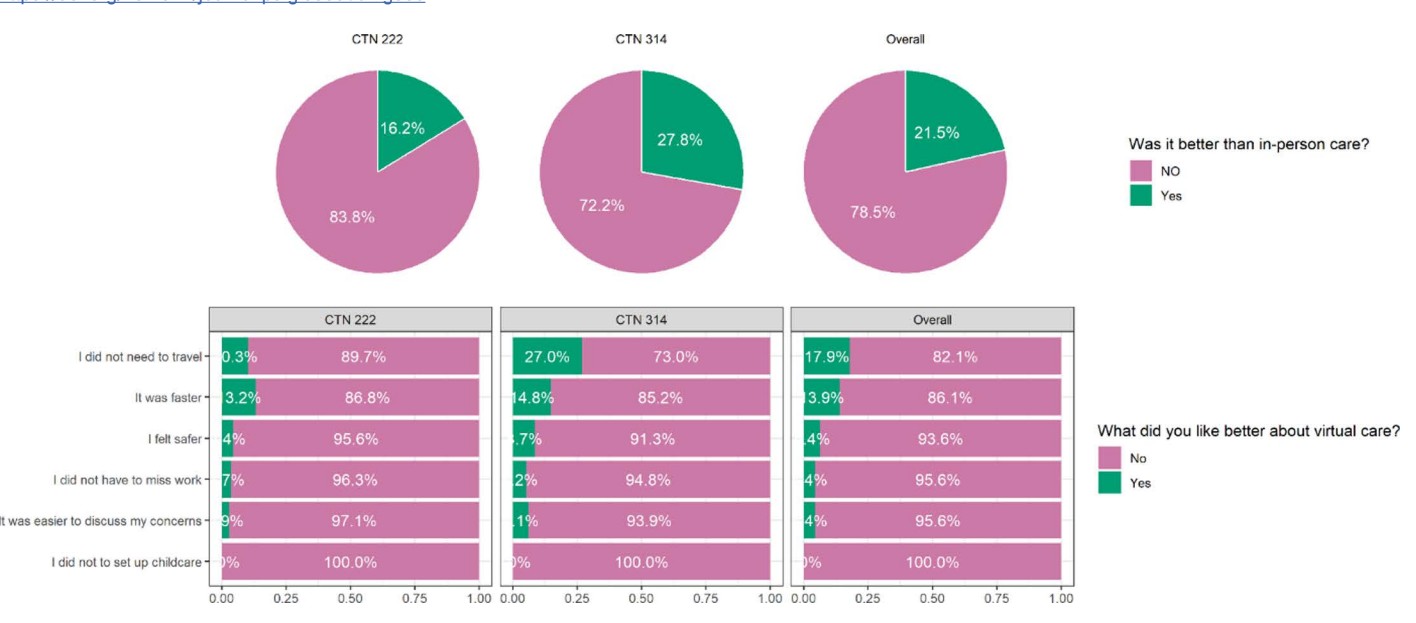

**Fig 4. Preference for virtual care.**

had not tried to access this service. This may in part been driven by financial considerations as physician compensation for virtual visits did require approval and extra paper work in the early days of the pandemic [9]. This observation does call for the need for clinics to ensure all clients have the knowledge and ability to access necessary visits.

On univariate assessment, those with a detectable HIV viral load at baseline were more likely to access virtual care. It may be that providers were concerned about those with detectable viral load and insisted on in- person care to address

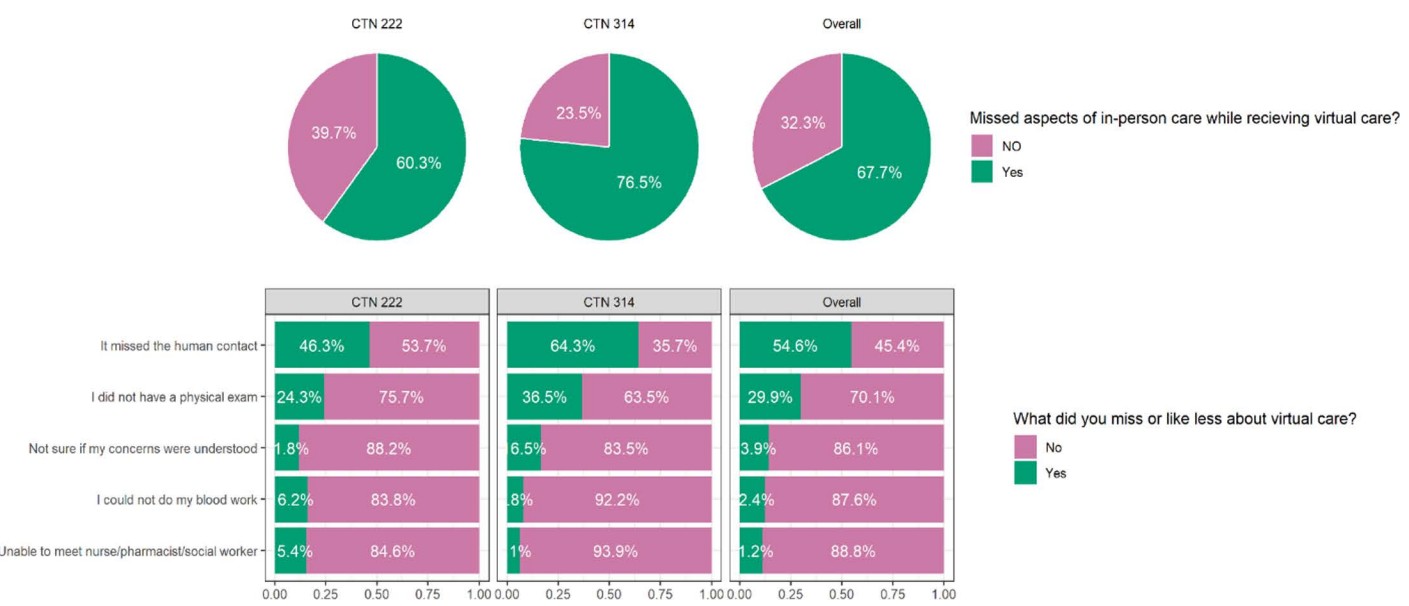

**Fig 5. Concerns about the virtual care experience.**

issues of viral failure or adherence. However, it raises concern that individuals with suboptimal adherence to therapy may not take sufficient care of their own health and be lost in a system that does not provide in person care, where the multidisciplinary team including nurses, and pharmacists could provide additional support.

When virtual care was accessed in our study, it was most typically with their primary care physician or with the HIV specialist. Utilization by other allied health services such as nursing and social work was much lower. This may have been due to restrictions on overall services offered during the pandemic and/or with reallocation of these services to other activities; especially in hospital settings.

When virtual services were used, more than 90% of our participants were satisfied to very satisfied with the experience, and less than 15% felt their clinical needs had not been addressed. Despite this level of satisfaction, only 20% felt virtual care was better than in person care citing less travel and time as the major advantages. In contrast, two-thirds missed aspects of in person care; primarily the personal contact. When the participants returned to clinic, many of the older group described loneliness and isolation and felt that the clinic provides an important contact for social wellbeing.

Despite the challenges of virtual care, the vast majority of our participants were able to conduct HIV specific blood work on a regular schedule and had their medication renewed on schedule. Both of these cohorts are engaged in care by virtue of their main study participation and may not reflect the general population of persons living with HIV.

Although our study revealed high satisfaction with virtual health our participants still preferred in patient contact. Access could still be an issue as in our study, 68% had annual incomes < $50K and 17% did not have access to a device allowing for virtual care. This was more prevalent in those with HCV coinfection.

The literature is varied on the use of and satisfaction with virtual care in persons living with HIV. Similar to our study, a survey of 202 patients followed at San Francisco-based HIV clinic, 11% preferred telehealth, 47% in person visit and 42% liked them equally [10]. A study of 371 persons living with HIV in Texas, 57% preferred telehealth and 37% stated that they would continue to use this modality after the pandemic [11].

The mode for virtual care delivery (i.e., telephone vs video) may be important. In a qualitative study with 31 persons living with HIV (median age 50 years, majority Latino/Hispanic of Black African-American) in Los Angeles [12], all felt

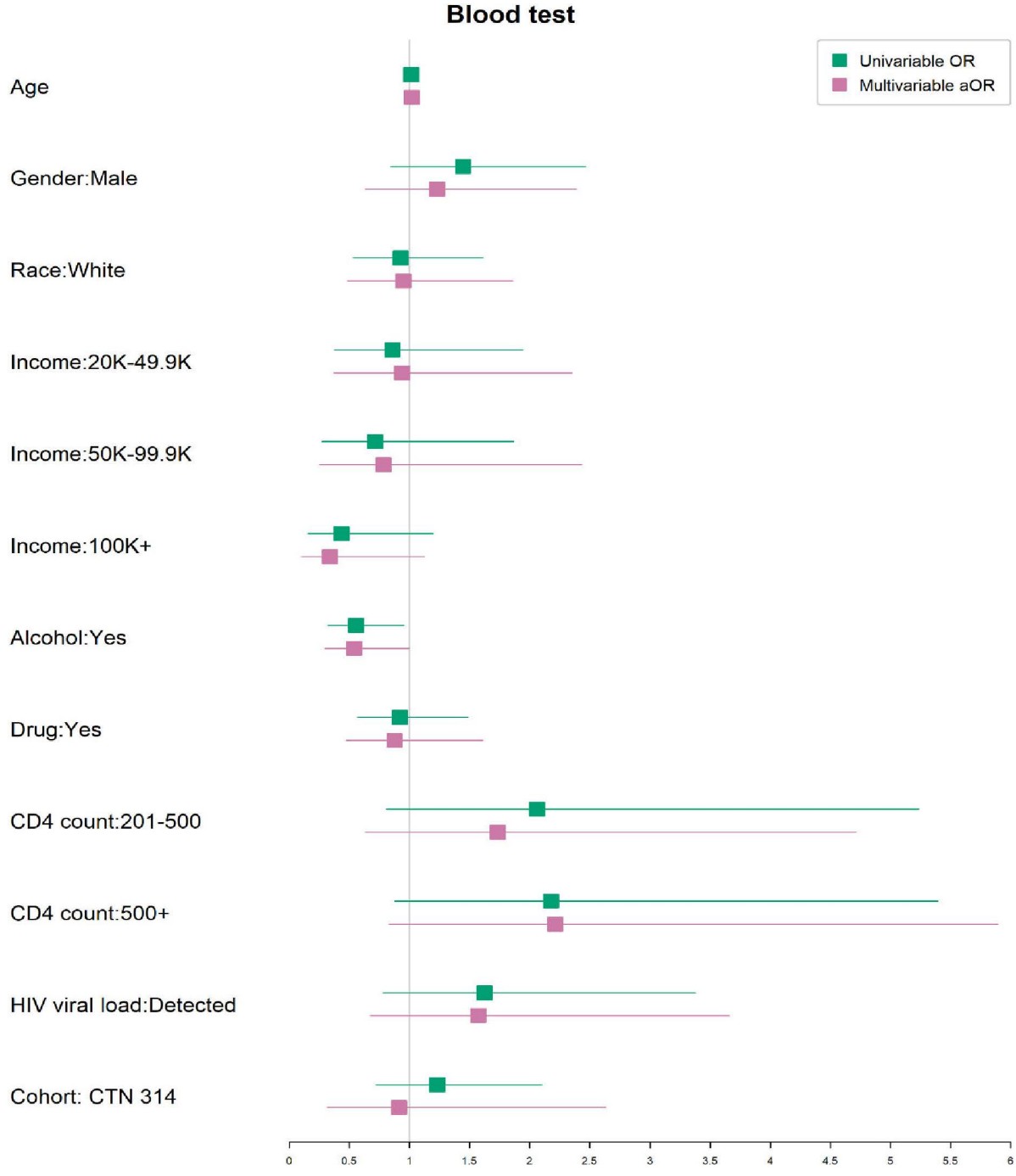

**Fig 6. Univariate and multivariate analysis of factors associated with the ability to complete regular laboratory testing during the COVID pandemic.**

capable of participating in telephone visits and nearly all wanted to continue telemedicine as part of their routine care. The main reason cited for the preference for virtual care was the time saved by not need to commute to clinic. They had been on ART an average of 8 years and felt their interpersonal relationship and dynamic with their clinician had been well-established and unaffected by virtual visits.

A study of 205 persons amongst two HIV clinics in the US Northeast (mean age 52 years) determined that of the 75% that had participated in virtual care, 42% perceived telemedicine visits (conducted more commonly by telephone than video) as useful during the pandemic [13]. Twenty seven percent thought that telemedicine would be useful after the pandemic; especially in the older age group.

A cross-sectional analysis of electronic health records from an urban HIV clinic housed in a large academic system in Connecticut assessed 1742 visits (1432 by telephone and 310 by video) [14]. Older age and Black race were associated with increased telephone visit utilization. Only 20% were over the age of 65 years. This research team cautioned that failure to provide telephone visits could result in the exclusion of vulnerable populations from care.

In a review of telehealth utilization in HIV care delivery, only 22% of HIV care providers in the US used telehealth for HIV care before the pandemic but this increased to 99% during the pandemic [8]. They observed that patients with telehealth visits were less likely to miss appointments and that several studies show high satisfaction rates with more than 80% willing to continue to use telehealth. The main advantages were convenience, flexibility, and reduction of HIV related stigma. They note several unresolved challenges including broadband access. They caution about disparities in access based on age, race, income and education.

Use and satisfaction of virtual care has been studied in other health care and disease areas in Canada and elsewhere. In a study of a Toronto ambulatory hospital, patient experiences with virtual care were generally positive but provider experiences less so [15]. A study evaluating persons self-isolating with COVID infection found that patients viewed virtual care positively but access to technology impacted the experience and satisfaction [16]. A discreet choice experiment involving physicians and patients from a primary care setting found that during the pandemic the most important considerations for virtual care vs in person visits were availability, time until the appointment, severity of the medical problem, patient-physician relationships and the flexibility of reception hours [17]. In a mixed methods study of 1226 patients referred to medical centers and hospitals in Iran, 71% preferred in person visits and 29% telemedicine. The primary telemedicine reasons were avoiding infectious diseases, saving costs, eliminating travel time. The primary in person visits were more accurate disease assessment, diagnosis and therapy. In a retrospective cohort study of primary care providers in Manitoba Canada, approximately 1/3 of 146, 372 visits were conducted virtually between March and June 2020. Females, those with more than 3 comorbidities and those with more than 10 prescriptions were more likely to have had at least one virtual visit [18]. In a study of 520 patients from 10 neuromuscular clinics 50% preferred in patient visits with the major concern about the lack of a physical exam [19].

All of these studies point to advantages and disadvantages of virtual care and to the need for individualization of models of care that reflect the population and individuals receiving care. Prior to the COVID pandemic, virtual care was seldom used [9] but research has shown this has increased post pandemic and during the recovery phase the balance between virtual and in person visits is reverting [20]. This has policy implications including the need for virtual care billing codes reflecting the services delivered, guidelines to seek and provide virtual care [2] and the need for digital innovation [1]. In the province of Ontario, 2.2-4.6 million persons do not have a primary care provider. Virtual care could improve the efficiency of the system and access to health care [21].

Models for virtual care [22] and newer modalities with mobile applications continue to be developed [23]. Payers need to develop a system to ensure appropriate physician compensation for telephone and video visits. Systems need to be in place to ensure appropriate monitoring of laboratory work, parameters associated with comorbidity outcomes, and timely renewal of prescription medications. More research is necessary to evaluate the long-term impact of virtual care on HIV and non-HIV related outcomes taking into consideration the many disparities which exist in this population. Globally, many persons even in resource limited settings have access to a mobile phone. WelTel Kenya1 was a randomized clinical trial of HIV infected persons initiating anti-retroviral therapy. Weekly mobile phone short message communication was associated with significantly improved adherence to therapy and rates of viral suppression relative to standard of care [24]. Similarly, PrEPmate a youth-tailored bidirectional text-messaging intervention in Chicago resulted in improved PrEP (pre-exposure prophylaxis for HIV) and adherence to study visits [25].

The strengths of our study include a large sample size and the inclusion of two key vulnerable populations living with HIV. Despite concerns about the ability to use virtual devices our aging cohort was older than those previously reported and yet still had access and use of the appropriate technology. As our cohort was primarily white, and male, we were unable to assess the impact of race or gender. As our participants were largely engaged in care and actively enrolled in ongoing cohort studies, the results may not be generalizable to all persons living with HIV in Canada nor applicable to other healthcare settings outside of Canada where access to technology could vary. We did not capture the proportion of our participants who had an in person visit during the same time period to determine the reasons behind some of the associations we observed. Further, we did not collect data on the specific virtual services were used by participant or provider nor on other aspects of care beyond scheduled appointments. We did not address the specific question of privacy and the impact that it could have on the use of virtual technology.

Our study demonstrated that virtual care was accessed by a high proportion of persons living with HIV during the COVID pandemic and more likely in our aging than participants from our HCV co-infected cohort. Although most were satisfied with the experience, most did not wish to continue with virtual care, a finding that is consistent with other studies. Clinics need to adapt their practices and consider hybrid models wherein individuals needs and preferences for virtual care and its modalities be considered. As the main reason for not accessing virtual care was that it was not offered, clinics must develop mechanisms wherein their clients can understand how and what means they are able to receive virtual care. Providers also need to support patient use of the technology and resolve related issues. Future research could evaluate prospectively, the preferences of virtual vs in person care and be focused on the particular issues- i.e., mental health, vs comorbidity vs HIV management. Further, there needs to be longer term studies that determine whether long term outcomes such as viral load suppression or development of viral resistance is impacted by increased use of virtual care. Future studies could also explore different telehealth reimbursement models that optimize provider participation and patient access to virtual care. This also needs to be extended to members of the health care team such as nurses, social workers and pharmacists.

We identified some challenges of virtual care for vulnerable populations. New virtual care models could involve multi-disciplinary teams to deal with the issues facing these groups. With this in mind, our group has been developing in collaboration with patients virtual models of geriatric care in HIV that we hope to implement in our clinics [26,27].

## Supporting information

**S1 Data. Virtual care questionnaire data.**
(XLSX)

## Acknowledgments

The CTN-COVID Sub-study Group also includes: Alexander Wong[8], Shariq Haider[9], and Pierre Cote[10]

[8]Department of Medicine, University of Saskatchewan, [9]Department of Medicine, McMaster University, Ontario, [10]Clinique Medicale du Quartier Latin, Faculty of Medicine, University of Montreal.

## Author contributions

**Conceptualization:** Sharon Walmsley, Marina B Klein.

**Data curation:** Bryan Boyachuk, Pamela Aldebes.

**Formal analysis:** Majid Nabipoor.

**Funding acquisition:** Sharon Walmsley, Marina B Klein.

**Investigation:** Valerie Martel-Laferriere, Mona Loutfy, Curtis Cooper, Marie-Louise Vachon.

**Writing – original draft:** Sharon Walmsley.

**Writing – review & editing:** Sharon Walmsley, Majid Nabipoor, Valerie Martel-Laferriere, Mona Loutfy, Curtis Cooper, Marie-Louise Vachon, Bryan Boyachuk, Pamela Aldebes, Marina B Klein.

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
