## [Decision Letter · Decision Letter 0]

10 Feb 2025

PDIG-D-25-00063Evaluation of the Virtual Care Experience for Persons in prospective cohorts with HIV and HCV coinfection and those with HIV older than 65 years of age during the COVID Pandemic.PLOS Digital Health Dear Dr. Walmsley, Thank you for submitting your manuscript to PLOS Digital Health. After careful consideration, we feel that it has merit but does not fully meet PLOS Digital Health's publication criteria as it currently stands. Therefore, we invite you to submit a revised version of the manuscript that addresses the points raised during the review process. Please submit your revised manuscript within 60 days Apr 11 2025 11:59PM. If you will need more time than this to complete your revisions, please reply to this message or contact the journal office at digitalhealth@plos.org. Please include the following items when submitting your revised manuscript:* A rebuttal letter that responds to each point raised by the editor and reviewer(s). You should upload this letter as a separate file labeled 'Response to Reviewers '. This file does not need to include responses to any formatting updates and technical items listed in the 'Journal Requirements' section below.* A marked-up copy of your manuscript that highlights changes made to the original version. You should upload this as a separate file labeled 'Revised Manuscript with Track Changes '.* An unmarked version of your revised paper without tracked changes. You should upload this as a separate file labeled 'Manuscript '. If you would like to make changes to your financial disclosure, competing interests statement, or data availability statement, please make these updates within the submission form at the time of resubmission. Guidelines for resubmitting your figure files are available below the reviewer comments at the end of this letter. We look forward to receiving your revised manuscript. Kind regards, Haleh AyatollahiSection EditorPLOS Digital Health Haleh AyatollahiSection EditorPLOS Digital Health Leo Anthony CeliEditor-in-ChiefPLOS Digital Healthorcid.org/0000-0001-6712-6626 **Journal Requirements:**1. Please provide an Author Summary. This should appear in your manuscript between the Abstract (if applicable) and the Introduction, and should be 150–200 words long. The aim should be to make your findings accessible to a wide audience that includes both scientists and non-scientists. Sample summaries can be found on our website under Submission Guidelines:

https://journals.plos.org/globalpublichealth/s/submission-guidelines#loc-parts-of-a-submission. **Additional Editor Comments (if provided):****Reviewers' Comments:** Reviewer's Responses to Questions

**Comments to the Author**

1. Does this manuscript meet PLOS Digital Health’s publication criteria ? Is the manuscript technically sound, and do the data support the conclusions? The manuscript must describe methodologically and ethically rigorous research with conclusions that are appropriately drawn based on the data presented.

Reviewer #1: Yes

Reviewer #2: Yes

2. Has the statistical analysis been performed appropriately and rigorously?

Reviewer #1: Yes

Reviewer #2: Yes

3. Have the authors made all data underlying the findings in their manuscript fully available (please refer to the Data Availability Statement at the start of the manuscript PDF file)?

Reviewer #1: Yes

Reviewer #2: Yes

4. Is the manuscript presented in an intelligible fashion and written in standard English?

Reviewer #1: Yes

Reviewer #2: Yes

5. Review Comments to the Author

Reviewer #1: This manuscript presents an important evaluation of virtual care experiences among persons ‎living with HIV. The study is methodologically sound and structured, with hypothesis, ‎appropriate statistical analyses, and a good discussion of findings.‎

All areas require clarification and revision to enhance the manuscript are listed in details ‎‎(please refer to the attached pdf file).‎

Reviewer #2: The authors present an observational cohort study that examines the impact of virtual care on vulnerable populations living with HIV in Canada during the COVID-19 pandemic. The authors surveyed participants from two established cohorts, CTN 222 (HIV/HCV coinfection) and CTN 314 (older than 65 years). They found that 55.3% of participants engaged in virtual care, with higher engagement in the older cohort. Satisfaction with virtual care was high, with 55% very satisfied, but 80% preferred in-person visits. The study highlights the need for flexible care models to meet the preferences and needs of patients.

I found the study and its findings interesting, important and presented clearly. I especially liked the clear presentation in the pie graphs, horizontal bar graphs, and colour schemes.

My main concern is definition of virtual care which they simply describe as “audio or video” without further definition or details in the manuscript itself. I later found this in the Supplementary material “Virtual care is when you have your appointment with the doctor/healthcare provider over the phone or the computer”. This is important because if any phone/computer/remote interaction is included in virtual care outside of an ‘appointment’ it would change the stats, findings, and perspective (see below).

I suggest also indicating which specific virtual services were used. For example, on the healthcare provider side, what audio and video systems/products/services were used? This could affect how often and in which ways virtual care was offered to this population (e.g. line 99-100), and ‘not being offered’ was a key reason for reporting non-usage of virtual care by the patients.

Similarly, on the patient side, their interpretation of what ‘virtual care’ means could influence their response. If virtual care does not include other times a clinician or healthcare team member provides aspects of care beyond scheduled appointments. This might especially affect whether they considered virtual care (encounters?) with allied health members such as nurses.

In Figure 1, ‘not having a computer’ was a reason for not participating in virtual care, but virtual care could include audio in which case only a phone may be required. Many virtual care platforms now include multi-modality communication types – video, phone/audio, and text messaging or secure messaging via patient portals or even remote patient monitoring via connected medical devices.

My other key recommendation is to compare the virtual care usage in these cohort populations with other general and similar non-HIV populations. There are published data on general population virtual care usage in Canada over the same time periods which would inform a comparative benchmark rather than just against other HIV cohort studies. This could at least be done in the discussion if not the results.

Additional minor points:

Line 53 Introduction – states physical exams can not be done by virtual care – but inspection can often be done by phone or video, as can assessing a patients voice. Some health programs have even developed standards for guided physical exams by virtual care.

Line 56 Introduction – states ‘privacy of home’ as a reason, but may exclude similar privacy consideration for homeless people or those at work or other locations.

Line 70 Introduction – does ‘face to face’ exclude video?

Line 133 and the Discussion – does human contact mean in person? This may depend on the type of virtual care. Other studies have reported that ongoing texting conversations have strengthened healthcare provider-patient relationships as they can be more responsive than booked appointments.

Line 158 Discussion – what virtual platform? – describe in the methods or findings

Line 162, 208-210 Discussion – compare also to non-HIV population literature (general, and perhaps HCV/addictions and other aging disease cohorts)

Line 182 Discussion – could it be that patients with higher viral loads experience more ‘virtual care’ because they were called about these results or more effort was made to book/follow them up?

Line 189 Discussion – see above re allied health and spontaneous/unscheduled phone calls

Line 213 Discussion – see above about other modalities of virtual care. OK if for this paper it is clearly defined as audio and video – but perhaps the discussion can mention other types

Line 251 – Discussion – why isn’t data available on in person visits? That would be helpful (though not critical)

Finally, I didn’t see any questions about privacy concerns from the patient survey. This could be mentioned as a limitation or discussed. That said, privacy is not usually a concern for patients as much as virtual care administrators tend to portray it. In fact, it can improve comfort with privacy to not have to sit in an HIV clinic waiting room where they might be seen by people they know.

6. PLOS authors have the option to publish the peer review history of their article (what does this mean? ). If published, this will include your full peer review and any attached files.

**Do you want your identity to be public for this peer review?** For information about this choice, including consent withdrawal, please see our Privacy Policy .

Reviewer #1: **Yes: ** Dr. Mohammed Sallam

Reviewer #2: **Yes: ** Richard T Lester

---

## [Decision Letter · Decision Letter 1]

14 Apr 2025

Evaluation of the Virtual Care Experience for Persons in prospective cohorts with HIV during the COVID Pandemic.

PDIG-D-25-00063R1

Dear Dr. Walmsley,

We are pleased to inform you that your manuscript 'Evaluation of the Virtual Care Experience for Persons in prospective cohorts with HIV during the COVID Pandemic.' has been provisionally accepted for publication in PLOS Digital Health.

Best regards,

Haleh Ayatollahi

Section Editor

PLOS Digital Health

**Additional Editor Comments (if provided):**

**Reviewer Comments (if any, and for reference):**

Reviewer's Responses to Questions

**Comments to the Author**

1. If the authors have adequately addressed your comments raised in a previous round of review and you feel that this manuscript is now acceptable for publication, you may indicate that here to bypass the “Comments to the Author” section, enter your conflict of interest statement in the “Confidential to Editor” section, and submit your "Accept" recommendation.

Reviewer #1: All comments have been addressed

Reviewer #2: All comments have been addressed

2. Does this manuscript meet PLOS Digital Health’s publication criteria ? Is the manuscript technically sound, and do the data support the conclusions? The manuscript must describe methodologically and ethically rigorous research with conclusions that are appropriately drawn based on the data presented.

Reviewer #1: Yes

Reviewer #2: Yes

3. Has the statistical analysis been performed appropriately and rigorously?

Reviewer #1: Yes

Reviewer #2: Yes

4. Have the authors made all data underlying the findings in their manuscript fully available (please refer to the Data Availability Statement at the start of the manuscript PDF file)?

Reviewer #1: Yes

Reviewer #2: Yes

5. Is the manuscript presented in an intelligible fashion and written in standard English?

Reviewer #1: Yes

Reviewer #2: Yes

6. Review Comments to the Author

Reviewer #1: Thank you for addressing and answering all the review points. Your efforts in revising the manuscript are appreciated. The revised version has shown improvements in several areas.

Reviewer #2: The authors have addressed my main concerns and recommendations.

I was unable to find the tracked changes or point by point response which made it more difficult to assess. But looking it over they seem sufficient.

Thank you

7. PLOS authors have the option to publish the peer review history of their article (what does this mean? ). If published, this will include your full peer review and any attached files.

**Do you want your identity to be public for this peer review?** For information about this choice, including consent withdrawal, please see our Privacy Policy .

Reviewer #1: **Yes: ** Dr. Mohammed Sallam

Reviewer #2: **Yes: ** Richard T Lester
